# The Evaluation of Selected Trace Elements in Blood, Serum and Blood Cells of Type 2 Diabetes Patients with and without Renal Disorder

**DOI:** 10.3390/nu16172989

**Published:** 2024-09-04

**Authors:** Marcin Kosmalski, Rafał Frankowski, Joanna Leszczyńska, Monika Różycka-Kosmalska, Tadeusz Pietras, Iwona Majak

**Affiliations:** 1Department of Clinical Pharmacology, Medical University of Lodz, 90-153 Lodz, Poland; marcin.kosmalski@umed.lodz.pl (M.K.); tadeusz.pietras@umed.lodz.pl (T.P.); 2Students’ Research Club, Department of Clinical Pharmacology, Medical University of Lodz, 90-153 Lodz, Poland; rafal.frankowski@stud.umed.lodz.pl; 3Institute of Natural Products and Cosmetics, Department of Biotechnology and Food Sciences, Lodz University of Technology, 90-537 Lodz, Poland; joanna.leszczynska@p.lodz.pl; 4Department of Clinical Electrocardiology, Medical University of Lodz, 92-213 Lodz, Poland; monika.rozycka-kosmalska@umed.lodz.pl; 5The Second Department of Psychiatry, Institute of Psychiatry and Neurology in Warsaw, 02-957 Warsaw, Poland; 6Institute of Food Technology and Analysis, Department of Biotechnology and Food Sciences, Lodz University of Technology, 90-924 Lodz, Poland

**Keywords:** type 2 diabetes mellitus, chronic kidney disease, micronutrients, chromium, nickel, manganese, zinc

## Abstract

Background: An appropriate diet is the basis for the treatment of type 2 diabetes (T2DM). However, there are no strict recommendations regarding the content of micronutrients and their modifications in the presence of chronic kidney disease (CKD). Therefore, we decided to investigate whether T2DM patients, including those with CKD, have different levels of chromium, nickel, cobalt, magnesium, and zinc in various blood elements compared to healthy individuals. Methods: We divided our subjects into three groups: the control group (individuals without T2DM and proper renal function), those with T2DM and proper renal function, and those with T2DM and GFR < 60 mL/min/1.73 m^2^. Results: We observed higher levels of chromium in all materials examined in patients with T2DM and impaired renal function. Both study groups found higher levels of nickel in samples of whole blood and red blood cells. Patients with T2DM and proper renal function had higher levels of serum manganese. Both study groups had lower levels of serum zinc. We observed higher levels of chromium in all materials examined in patients with T2DM and impaired renal function. Both study groups found higher levels of nickel in samples of whole blood and red blood cells. Patients with T2DM and proper renal function had higher levels of serum manganese. Both study groups had lower levels of serum zinc. Conclusions: In order to ensure effective care for patients with T2DM, it is necessary to improve the standard diet, including the content of micronutrients and their modification in patients with concomitant CKD.

## 1. Introduction

Nowadays, medicine is dealing with an increasing prevalence of diabetes mellitus. Data show that 10.5% of the adult population suffers from this disease, of which the most common type is type 2 diabetes mellitus (T2DM) [1]. Lifestyle properties, including eating habits, are among the factors that lead to this condition [2]. What is in line with this modification of lifestyle and proper dietary interventions are the cornerstone of T2DM treatment [3]. We have data showing that patients with T2DM have abnormal micronutrient content than those without T2DM [4,5]. Disbalance in nutrient levels has an influence on T2DM pathogenesis, such as glucose homeostasis and insulin resistance [6]. Research has demonstrated a correlation between the levels of trace elements and the risk and pathogenesis of T2DM [7]. T2DM leads to many complications, including chronic kidney disease (CKD) and cardiovascular problems [8]. A proper and balanced diet has a significant impact on human body functioning and can prevent disease development and premature mortality [9]. Studies indicate that chosen dietary patterns can be preventive for T2DM development [10]. What is more, available data indicates the positive effects of nutritional interventions in the management of patients with diabetes [11]. This data demonstrates the importance of dietary interventions, even in conditions that occur before T2DM onset. There is a lack of data indicating dietary recommendations for people with diabetes regarding micronutrient content, especially for those with coexistent CKD. It should be added that obtaining such information is a clinically important matter because it is necessary to know what diet to recommend to a person with T2DM and a person with T2DM and CKD.

The objective of this study is to assess the levels of specific micronutrients in the blood, serum, and red blood cells of T2DM patients who have CKD or normal kidney function. The acquired data will enable the implementation of appropriate nutritional interventions to enhance the overall well-being of individuals with diabetes.

## 2. Materials and Methods

The study included a total of 90 patients (36 males and 54 females) between the ages of 41 and 89 (median 68.05 ± 12.70 years) who were hospitalized in the Department of Internal Medicine, Diabetology, and Clinical Pharmacology of the Regional Specialist Hospital in Zgierz, Poland, between 2010 and 2012. All patients provided written informed consent. The Medical University of Lodz Committee on the Ethics of Research in Human Experimentation (approval number RNN/182/17/KE, obtained on 15 December 2009) approved the study, which was conducted in accordance with the Helsinki Declaration for human research.

The division into categories was conditioned by the absence of glucose metabolism disorders or the diagnosis of T2DM, which served as the inclusion criteria. The control group consisted of people without chronic diseases, acute conditions and with applied diagnostic methods that may affect the value of micronutrient determination (people with such conditions were not included in the study). The study did not include patients with impaired glucose tolerance or anomalous fasting blood glucose. In addition, people who used medications or dietary supplements that could affect the micronutrient determination were not included in the study. The subsequent phase involved the assessment of participants for CKD. The diagnosis of CKD was made on the premise of a glomerular filtration rate (GFR) that was less than 60 mL/min/1.73 m^2^. Patients who were diagnosed with CKD but did not have T2DM were also excluded from the study. The diagnosis of T2DM was based on the current diagnostic criteria of the Polish Diabetes Association and the patient’s medical history.

Consecutively, each of the designated patients was assigned to one of three categories. The first trial population comprised patients with T2DM and CKD, the second with T2DM and adequate renal function (GFR > 60 mL/min/1.73 m^2^), and the third of healthy subjects without T2DM and CKD. Efforts were made to ensure that each cohort had an equal number of men and women. The low representation of men, particularly in the group with the most advanced disease, among the total patients, made this task quite challenging. The patient distribution among the three groups is as follows: the first group (T2DM and CKD) consists of 22 women and 8 men, the second group (T2DM and adequate renal function) of 17 women and 12 men, and the control group of 16 women and 16 men.

Patients included in the study followed up on dietary management recommended in accordance with the dietary standards for the Polish population [12]. Each patient was required to maintain a daily diet diary for a minimum of three days prior to the biological material collection for testing. In accordance with the “minimal” dietary interview methodology, patients were also advised to include at least one weekend day and two weekdays. Patients were also asked to provide precise information regarding their medications and dietary supplements.

Blood pressure (RR), body mass index (BMI), and waist–hip ratio (WHR) were determined by obtaining anthropometric measurements, including weight, height, waist, and hip circumference, after the specified number of days.

Standard blood laboratory tests were administered to each patient to evaluate renal function (creatinine and urea levels), carbohydrate metabolism balance (fasting glucose—FG, percentage of glycated hemoglobin—HbA1c), lipid metabolism balance (total cholesterol—TCH, low-density lipoprotein—LDL, high-density lipoprotein—HDL, and triglycerides—TG), liver function (activity of alanine aminotransferase—ALT, aspartate aminotransferase—AST, gamma-glutamyl transpeptidase—GGTP, and total bilirubin concentrations), uric acid, potassium, sodium, and basic morphological parameters were determined.

The samples were collected in 7.5 mL Monovette Li-Heparin closed-system tubes (Sarstedt), (SARSTEDT AG & Co. KG Sarstedtstraße 1 51588 Nümbrecht Germany) which were coated with lithium heparin. Together with the samples, a standard reference serum (Seronorm) (SERO Stasjonsveien 44-1396 Billingstad Norway) was transferred manually to Monovette tubes and processed with samples. After mineralization in 65% Merck Millipore Suprapur nitric(V) acid, (Merck KGaA Frankfurter Straße 250 Darmstadt, 64293, Germany), the samples were diluted to a total volume of 25 mL with hplc-grade water. Subsequently, the samples were divided into two equal portions. One was frozen at −80 °C for further investigation, while the other was centrifuged for 5 min at approximately 2000 (or 5000) rcf within an hour of collection. A minimum of 2 mL of plasma was separated and stored for future research. In order to prevent the disruption of minerals from the cell and their release into the plasma, the plasma was centrifuged within one hour of collection. For this objective, a centrifuge program that decelerates slightly was implemented. The plasma was subsequently separated and transferred to a separate tube with a minimum of 2 volumes. For additional analysis, a vial containing 2 mL of plasma and another containing at least 2 mL of whole blood were employed.

The matrix effect was subsequently eliminated by utilizing an Ertec Magnum II microwave mineralizer to perform pressure microwave digestion of 1 milliliter of whole blood and 1 milliliter of plasma in 5 milliliters of spectrally pure nitric acid (V). The mineralizate was prepared at the appropriate dilutions for the mineral to be determined. An atomic absorption spectroscopy apparatus was employed to generate a standard curve using standard solutions. In this investigation, an atomic absorption spectroscope model 932 plus from GBC was implemented. The monochromator had a resolution of 0.2 nm and a wavelength range of 185 to 900 nm. The dual beam optical system was capable of correcting background and flame emissions in this instance by employing asymmetric modulation with a measurement interval of 0.01 nm. Detailed assay parameters are shown in Table 1. Reference materials that were certified by Merck AAS Certipur were implemented for calibration purposes. Standards from Merck were AAS standards for respective metal ions diluted into the same batch of Suprapur nitric acid (Merck). A blank sample was prepared with hplc-grade water in place of serum, then was mineralized and diluted along with the samples. The AAS result of this sample was deducted from other results to correct for the chemical impurities in the chemical reagents used.

Atomic absorption spectroscopy was employed to directly determine the concentrations of the selected ion in plasma and whole blood samples. The selected mineral’s intracellular content was determined by subtracting its concentration in whole blood from its concentration in plasma and then adjusting for hematocrit value.

The metal ions were determined using the ICP-MS method in this investigation. Colorimetric methods are typically employed to determine the metal ion content of blood or plasma samples. These methods involve quantifying the absorbance of complexes of metal ions with a variety of ligands. Complex compounds with high stability are employed in colorimetric procedures that are employed in routine determinations. In the event of acidosis, the cell may produce a variety of compounds that can function as ligands, such as β-hydroxybutyric acid and acetoacetate, if there is an absence of adequate metabolic equilibrium. These compounds possess robust complex-forming capabilities and can establish more robust complexes with the metal ions that are being labeled than with the ligands employed for colorimetric determination. Stronger complexes are generated with citrates and oxalates in the case of sodium azide. In such instances, the results may be inadequate to the actual concentration of the analyte being determined, i.e., they may be underestimated. In diabetes mellitus, the likelihood of the formation of a variety of compounds in the cell that are the result of defective metabolism (e.g., glucose, fatty acids) is high, and acidosis may develop, even if it is not clinically apparent.

In order to check the validity of the determinations carried out, and the reliability of the results obtained, determinations of chromium, manganese, cobalt, nickel, and zinc in the reference serum were carried out according to the procedure described.

For chromium, 4.20 ± 0.11 µg/L, manganese 19.0 ± 0.03 µg/L, cobalt 3.2 ± 0.18 µg/L, nickel 9.2 ± 0.31 µg/L were determined, while the declared contents are 4.8 µg/L, 19.9 µg/L, 3.2 µg/L, and 9.8 µg/L, respectively. The results obtained are within 95% agreement. Only slightly more divergent results were obtained for zinc. The determined zinc content of the reference serum samples tested was 2.742 ± 0.311 mg/L, while the declared concentration should be 2.440 mg/L. However, the discrepancies between these results are not too large and are within the upper limit of 95% acceptability. The analysis shows that the results obtained from the procedures used provide reliable values.

The Ertec Magnum II mineralizer manufacturer’s standard mineralization procedure was implemented. In terms of parameters, we used 1.0 mL of blood and plasma samples, each in 5 mL HNO3 (nitric acid), as a dissolver. The time was set as 7 min, and power was 100% for each. In both samples, the pressure range was set as 42–45 at. In a closed system, a single digestion reagent was employed to conduct a one-step digestion.

STATISTICA 12 Software (StatSoft Inc., Tulsa, OK, USA) was employed to conduct the statistical analysis. The mean values ± standard deviation were used to express the anthropometric and biochemical characteristics. The Mann–Whitney U test was employed to evaluate the differences between the two groups, as the distribution of variables did not conform to the normality established by the Shapiro–Wilk test. Statistical significance was defined as a *p*-value of less than 0.05.

## 3. Results

Control, T2DM with GFR greater than 60 mL/min/1.73 m^2^, and T2DM with GFR less than 60 mL/min/1.73 m^2^ were the three major groups chosen for statistical analysis. The categories were further divided and analyzed based on gender. Participants’ characteristics with comparisons between groups are presented in Appendix A. The recent study presents all results for the general categories, as no statistically significant differences between men and women were obtained. The BMI, total blood glucose, and HbA1 values of both T2DM patient categories are statistically significantly different from those of the control group. In addition, the T2DM GFR > 60 mL/min/1.73 m^2^ group exhibits substantially higher AST and ALT parameters than the control group, while the T2DM GFR < 60 mL/min/1.73 m^2^ group has higher uric acid, urea, and creatinine levels than the control group.

### 3.1. Results of Chromium Determination in Whole Blood, Serum, and Blood Cells

The body needs chromium, an element that is vital to the metabolism of carbohydrates. Concentrations of chromium are particularly important when it comes to diabetes, which is why this element has received special attention. In this investigation, we compared the concentrations of this micronutrient in healthy individuals with T2DM who had normal and impaired renal function, as well as in their whole blood, serum, and red blood cells. Table 2 presents the findings from these calculations.

All samples had chromium concentrations in the range of a few micrograms per liter. In the blood, the concentrations were higher compared to the serum, indicating that the chromium content in blood cells is slightly elevated compared to the serum. According to the study, there were disparities in chromium levels between healthy individuals and those with diabetes. Individuals with T2DM exhibit elevated levels of chromium in both their serum and whole blood. The maximum concentration of chromium in the blood is 9.55 ± 4.36 µg/L, while in the serum, it can reach as high as 11.98 ± 5.1 µg/L in individuals with impaired kidney function. It does not matter if we look at the differences between patients with a GFR below 60 or above. The differences are statistically significant at the 0.01 and 0.05 probability thresholds. Examining the chromium content in blood cells reveals similar correlations among the studied groups. The results are statistically significant at the probability levels of 0.01 and 0.05. Patients with impaired renal function have a significantly higher chromium content in their blood cells, with an average of 11.44 ± 3.62 µg/L, compared to healthy subjects who have an average of 6.52 ± 3.05 µg/L. When examining the results by gender [Table 2], it is evident that the blood chromium levels vary significantly between men (statistically significant at the 0.01 level) and women (statistically significant at the 0.05 level) in the group of patients with impaired renal function.

In both females and males, the elevated chromium content in red blood cells is the most pronounced and authoritative, as is the group with impaired renal function, which is statistically significantly different [Table 2 and Table 3]. The chromium content in this group of patients averages 10.54 µg/L in females and as much as 13.91 µg/L in males. Of note is the elevated plasma chromium content in diabetic women’s GFR < 60 of 13.75 µg/L, almost twice that of males. This elevated plasma chromium content in women may indicate an intake of dietary supplements containing this element, as women are more likely to take vitamin and mineral preparations.

### 3.2. Results of Nickel Determination in Whole Blood, Serum, and Blood Cells

The results of the study are very interesting because they indicate that the serum levels of this element do not differ significantly between healthy and sick individuals. Still, levels are notably higher in the whole blood of patients, particularly those with impaired kidney function [Table 2].

Sick patients’ red blood cells exhibit slightly higher levels of nickel. The concentrations of nickel in both the blood and red blood cells show a statistically significant difference at a probability level of 0.01 [Table 2].

The results of nickel content by gender are very interesting [Table 2].

The concentration of nickel in the serum of both diabetic and healthy females is significantly higher than in males, as shown in Table 2. In males, the concentration of nickel in the blood remains relatively stable, with a slightly lower level observed in patients. Women also exhibit this pattern. For whole blood determinations, the results are similar. Women with diabetes have a significantly higher nickel content compared to healthy women, and this content is also significantly higher than in males. These findings suggest that there is a higher availability of women and a higher uptake of nickel in women with T2DM. The concentration of nickel in erythrocytes most effectively demonstrates its presence in the body. Table 2 demonstrates a notable buildup of nickel in the red blood cells of female patients with healthy kidney function, with an average nickel content of 9.03 µg/L. Additionally, there is a significantly higher concentration of nickel in the red blood cells of both female and male patients with T2DM and GFR below 60, measuring 16.29 µg/L and 10.34 µg/L, respectively. Once again, the levels of nickel are considerably greater in women compared to men.

Although the negative effects of nickel on people with diabetes have not been confirmed, prophylactic recommendations to limit nickel in the diet should be considered.

### 3.3. Results of Cobalt Determination in Whole Blood, Serum, and Blood Cells

This study revealed that the cobalt levels in diabetics and the reference group were similar and did not significantly differ [Table 2]. This is applicable to all types of blood samples, including whole blood, plasma, and blood cell samples. The average serum cobalt concentrations were 1.2 µg/L, with patients having slightly higher levels at 1.42 µg/L, but the difference was not statistically significant. The levels of cobalt in red blood cells were considerably greater than in the blood serum, but there was no significant difference between the patient and healthy groups.

Analogous results were obtained in groups of subjects divided by gender [Table 2].

### 3.4. Results of Manganese Determination in Whole Blood, Serum, and Blood Cells

The findings show that manganese levels in blood cells are much higher than in serum. Manganese tends to accumulate inside the cell, resulting in a concentration of approximately 14–18 µg/L in serum and 90–105 µg/L in erythrocytes. Nevertheless, similar to cobalt, the analysis of manganese levels in the serum and whole blood of both healthy and diseased individuals revealed no significant variations, as indicated by the results presented in Table 2.

Serum manganese concentrations were significantly higher in patients with GFR < 60 compared to healthy subjects, but only at the 0.05 probability level [Table 2]. The results calculated by gender are identical [Table 2].

Only in the group of diabetic men with impaired renal function are the results statistically significant relative to healthy subjects at the 0.05 probability level [Table 2].

Perhaps changes in the concentration of manganese associated with superoxide dismutase or other forms of manganese responsible for scavenging free radicals are too small to be recorded.

### 3.5. Results of Zinc Determination in Whole Blood, Serum, and Blood Cells

The findings of our study validate the decrease in zinc levels observed in patients with T2DM whose serum zinc concentrations are lower compared to those of individuals without the condition. The average serum zinc content in healthy individuals is 3.544 mg/L. In patients with T2DM and GFR > 60, the serum zinc content is 1.641 mg/L, while in patients with T2DM and GFR < 60, it is 2.249 mg/L. Table 2 displays the comprehensive findings.

The serum zinc levels show statistically significant differences between the diabetic and healthy groups with GFR > 60 at a probability level of 0.01. Similarly, there are statistically significant differences between the group with GFR < 60 and the healthy group at a probability level of 0.05. These findings are presented in Table 2. The zinc concentration in serum and blood is marginally lower in patients with normal renal function compared to those with a GFR below 60, but this difference is not statistically significant. The zinc concentration in erythrocytes is significantly lower in individuals with diabetes, with an average range of 4.563 to 4.902 mg/L depending on GFR. In contrast, healthy individuals have a zinc concentration of 6.447 mg/L. This suggests a depletion of zinc reserves in the body.

The levels of zinc in the blood and erythrocytes of diabetic patients are generally lower than those of healthy individuals. However, these differences are not statistically significant, except for the concentration of zinc in the blood of patients with a GFR greater than 60 and healthy individuals. In this case, the difference is statistically significant at a probability level of 0.05. Serum determinations of zinc are both authoritative and sufficient for assessing the deficiency of this element in the body. Simultaneously, a deficiency of zinc primarily in the bloodstream may suggest a dysfunction in its absorption.

Table 2 displays the data on zinc content in sick and healthy subjects categorized by gender. The relationships between zinc content and gender were analyzed accordingly. When compiled in this manner, the results demonstrate statistically significant disparities in serum zinc levels between the groups of diabetic and healthy women and diabetic and healthy men. However, there are no statistically significant differences between genders within the same group [Table 2].

### 3.6. Results of Assessment of Elemental Contents in the Diet of Diabetic and Healthy People

Subjects suffering from T2DM consumed significantly higher amounts of zinc, manganese, and nickel than controls. However, due to residual data on chromium and copper content, it was not possible to determine these elements in patients’ diets [Table 3].

## 4. Discussion

To our knowledge, guidelines outlining the recommended daily consumption of trace elements in the diet for people with T2DM and CKD are lacking. Because there has not been much extensive clinical research on the subject, the majority of guidelines emphasize consuming fundamental nutrients and calorie-restriction diets without directly indicating the recommended daily intake of micronutrients [13,14].

Clinical recommendations for the management of people with diabetes issued by the Polish Diabetes Association and the American Diabetes Association (ADA) Standards of Care in Diabetes do not indicate the need for microelement supplementation in patients without a diagnosed deficiency; rather, they focus on a healthy diet with proper macronutrient intake, such as reducing carbohydrates and body weight. According to diabetics with CKD, only a reduction of sodium intake and a calorie-restricted diet with proper protein intake can be found [15,16].

It should also be emphasized that there is no universal diet for all people with diabetes. The diet should be adjusted to the patient, taking into account their comorbidities and the possibility of using a given type of diet.

We observed that our patients suffering from T2DM with or without CKD had higher dietary intakes of zinc, manganese, and nickel compared to healthy individuals. On the other hand, a study revealed a lower intake of zinc in T2DM patients with foot ulcers [17].

Differences in the content of individual elements in the diets of people with diabetes and healthy people indicate that the former consume more foods rich in zinc, manganese, and nickel. This elemental composition is characteristic of many plant products. However, the content of metal ions in plants is often inadequate for efficient absorption due to their fiber content and substances that impede their biological utilization. For example, although plant products nominally contain the most zinc, after considering the absorption capacity of these elements, the best sources are foods containing animal proteins.

According to the results obtained in this study, blood transition metals are predominantly cell-associated and lower than free metals in serum, which may refer to the results obtained in this study.

### 4.1. Chromium

Chromium’s role is critical because it plays a significant role in the metabolism of carbohydrates and lipids. Research has shown that the addition of chromium to one’s diet can enhance the body’s ability to respond to insulin, resulting in improved insulin sensitivity. Researchers have linked the development of T2DM to insufficient dietary intake of chromium [18,19,20]. Taking chromium supplements has been shown to lower HbA1c levels in people with T2DM, which means that the disease’s metabolism is better controlled [21]. Research has demonstrated that the level of chromium in the serum tends to decline as individuals get older [22]. Our study found that people with T2DM and a GFR lower than 60 had higher levels of chromium in their full blood, serum, and erythrocytes than healthy people. Considering sex differences, significant variations were observed in females with T2DM and GFR lower than 60 in serum and erythrocytes. In contrast, significant differences were found only in full blood samples from diabetic males with GFR lower than 60. This contrasts with other studies that reported decreased levels of chromium in the serum of T2DM patients [23,24]. Researchers conducting a study on whole blood samples also found reduced levels of chromium in individuals with diabetes [5]. Nevertheless, a study has demonstrated increased concentrations of chromium in the blood samples of non-smokers with diabetes. However, another study did not find any differences between groups in terms of chromium levels in blood samples [25]. Furthermore, research has demonstrated decreased plasma chromium levels in individuals with T2DM [4]. Moreover, this also applies to patients who are currently in the pre-diabetes stage [26]. Additionally, it was noted that individuals with T2DM had elevated levels of chromium in their plasma and blood components compared to the control group [27]. We observed significantly elevated chromium levels in both serum and blood cells in subjects with a GFR below 60 compared to those with a higher GFR. However, a study found no significant changes in chromium levels in the serum of T2DM patients with chronic kidney disease at different stages [28]. A study has also shown a reduction in chromium levels in individuals with diabetic nephropathy [29]. When considering gender, we found that females with T2DM and a GFR < 60 had significantly higher chromium levels in their serum compared to males, with the values in females being almost twice as high as those in males. In contrast, when examining individuals with T2DM and a GFR below 60, there were greater disparities observed in blood samples from males compared to females. We must emphasize that the patients’ dietary supplements could potentially influence our findings. In short, the different results found for the amount of chromium in samples from people with T2DM show the complexity of interactions within the human body. To establish definitive evidence, additional research must be conducted on larger and more diverse groups while considering the correlation between chromium levels and patient age.

### 4.2. Nickel

Despite the potential cytotoxic effect of nickel on pancreatic β-cells and its possible influence on the development of diabetes, there have been relatively few studies dedicated to investigating the role of nickel in diabetes and its concentration in physiological fluids [30,31]. Conversely, nickel may have a dual impact on the risk of T2DM and function as a protective agent [32]. Another potential role in diabetes development could be the induction of hyperglucagonemia and hypoinsulinemia caused by nickel administration [33]. Our study found no significant variations in serum nickel levels among the groups examined, with the exception of its lower serum levels in diabetic men. Nevertheless, we noticed increased nickel concentrations in both the full blood and erythrocyte samples of both diabetic groups when compared to the control group. Considering gender, statistically significant differences were found between women in this regard and men with normal renal function in the whole blood sample. Consistent with previous research, a study has found elevated levels of nickel in the blood samples of non-smoking patients with T2DM [34]. People with diabetes with a GFR below 60 exhibited higher values. Conversely, a study found lower levels of nickel in whole blood samples of patients diagnosed with T2DM [5]. A separate study found no significant disparities in the levels of nickel in blood samples between individuals with diabetes and those who are healthy [25]. When examining gender differences in serum and blood cells, we found that women had higher levels of nickel compared to men in all the groups studied. We discovered that women with diabetes had higher nickel levels than both women and men without T2DM. We observed elevated levels of nickel in the erythrocytes of women with diabetes and normal kidney function. Additionally, both males and females with a GFR below 60 showed elevated levels of nickel in their red blood cells. The diabetic men’s groups exhibited reduced levels of nickel in both serum and whole blood compared to the control group. It is advisable to take precautionary measures to restrict nickel consumption in the diet, even though the exact negative impacts of nickel on individuals with diabetes remain undetermined.

### 4.3. Cobalt

Cobalt is a crucial element in the prevention of cardiovascular complications, particularly in individuals with diabetes. It should be emphasized that these positive effects may be related to vitamin B12 (methylcobalamin), which is the principal form of cobalt known to be nutritionally useful. An element known as the protoporphyrin complex has the ability to reduce oxidative stress, a common occurrence in diabetes and other conditions [35]. It was observed that cobalt is related to T2DM [36]. Researchers have found that cobalt supplementation positively impacts glucose levels, albeit solely in rats [37]. There is a notable lack of research on the metabolic effects of cobalt on humans with diabetes. The results of our study did not show any significant variations in the levels of cobalt in the samples analyzed across all the groups studied, including differences between males and females. Consistent with another study, there were no significant differences in cobalt levels between non-smokers with T2DM and healthy individuals [34]. Conversely, a study found higher levels of cobalt in whole blood samples from patients with T2DM [38]. A study using a rat model of T2DM found that cobalt chloride supplementation had a renal protective effect. On the other hand, some forms of cobalt are known as carcinogens—those with cobalt (II) oxidation state. Accordingly, caution should be exercised when selecting supplemental forms of cobalt. The data above suggests that cobalt in the form of methylcobalamin may have potential as a therapy for individuals with diabetes [39]. However, there is a lack of extensive research on the impact of cobalt on disease progression.

### 4.4. Manganese

Research pertaining to diabetes has relatively neglected manganese despite its significant role in mitigating excessive oxidative stress and eliminating free radicals [40]. Existing literature contains limited studies that indicate a potential involvement of manganese in diabetes, specifically in the progression of complications. Nevertheless, research has demonstrated that consuming more manganese can act as a safeguard against T2DM [41,42]. Our study found that diabetic males with impaired kidney function had significantly higher levels of manganese in their blood serum. However, a meta-analysis has demonstrated that the levels of manganese in the serum or plasma of individuals with diabetes are lower [43]. The results from the other determinations for each group and sample did not exhibit any statistically significant disparities in manganese content. A study found elevated levels of manganese in the whole blood of individuals with T2DM. However, when analyzing our full blood samples, we did not observe the same difference [38]. Conversely, some researchers have noted decreased manganese levels in whole blood samples among individuals with diabetes. Furthermore, this study found that the levels of manganese in the bloodstream were lower in patients with renal dysfunction compared to those with normal kidney function [5,44]. A separate study found no significant difference in plasma and full blood magnesium levels between individuals with diabetes and those without the condition [45]. Based on the levels observed in samples from T2DM patients, further research is necessary to elucidate the role and mechanism of manganese in relation to T2DM. It is reasonable to assume that the oxidative stress observed in individuals with diabetes leads to increased levels of superoxide dismutase, an enzyme that contains manganese. As a result, concentrations of this element should be higher. Another possible explanation is the accumulation of manganese in the body as a result of impaired kidney function and the decreased capacity of nephrons to filter and concentrate urine.

### 4.5. Zinc

Zinc is a crucial element in relation to diabetes. Despite extensive research, the function of this component remains uncertain. Nevertheless, the majority of studies carried out and documented in the scientific literature indicate a decrease in the amount of zinc in the blood of individuals with diabetes. Additionally, these studies suggest that supplementing with zinc can have a positive impact on blood sugar levels and can help delay the onset of complications associated with diabetes [46,47].

Both studied groups exhibited significantly lower serum zinc levels compared to the control group. Other researchers have also demonstrated similar variations [43,45,46]. However, in our study, the differences were not statistically significant in the group of men with T2DM and a GFR below 60. Conversely, a study has found higher levels of zinc in whole blood samples from patients with T2DM [38]. Analysis of blood cells and whole blood samples from diabetics with a GFR below 60 did not show any notable distinctions between the groups.

The findings suggest that zinc supplementation is essential for individuals with T2DM, particularly because numerous studies confirm the beneficial and protective impact of this mineral [46,48].

### 4.6. Limitations

Our study has several limitations. First, the patients’ diets lack chromium and copper. Second, there is no breakdown of CKD patients by stage of kidney disease. Third, there are a relatively small number of male subjects in the group with T2DM and CKD compared to women.

## 5. Conclusions

Considering the significant public health risk posed by diabetes, both in terms of its widespread occurrence and the associated complications, it is imperative to establish dietary guidelines that can effectively prevent the onset of diabetes or the development of its complications. Our study revealed significant differences in selected trace elements in assessed samples of diabetics with and without CKD. Our results identify the following observable pattern: The analysis revealed higher levels of chromium in all materials examined in patients with T2DM and a GFR below 60. Both study groups found higher levels of nickel in samples of whole blood and red blood cells. Patients with T2DM and a GFR above 60 had higher levels of serum manganese. Both study groups had lower levels of serum zinc. As we mentioned in the discussion, data from studies in selected trace element determination in T2DM patients are inconsistent, and studies performed on larger subjects are needed to clear up any doubts in this matter. In summary, there is a need to introduce standardized dietary recommendations to diabetic patients and those at risk of developing diabetes in clinical practice. The presence of renal dysfunction should also be taken into account. Based on our research, we propose introducing dietary patterns in T2DM with a lower content of nickel and a higher content of zinc. If CKD is observed in this group of patients, the diet should be modified to include a lower content of chromium.

## Figures and Tables

**Table 1 nutrients-16-02989-t001:** Determinability of selected metal ions by ASA method (Seronom standard serum).

	Chromium [µg/L]	Manganese [µg/L]	Cobalt [µg/L]	Nickel [µg/L]	Zinc [mg/L]
Declared content in the reference serum	4.8	19.9	3.2	9.8	2.440
Determined content in the reference serum	4.20 ± 0.11	19.00 ± 0.03	3.2 ± 0.18	9.2 ± 0.31	2.742 ± 0.311
Range of 95% compliance	4.4–5.2	18.8–21.0	3.0–3.4	9.2–10.4	2.107–2.773
Acceptable range	4.0–5.6	17.7–22.1	2.8–3.6	8.6–11.0	1.774–3.106

**Table 2 nutrients-16-02989-t002:** Elements contents in serum, whole blood, and erythrocyte samples of diabetic and healthy subjects, differentiated by gender and GFR parameter.

Element		Sample	Control	T2DM and GFR > 60	T2DM and GFR < 60
**Chromium [µg/L]**	**All**	Whole blood	6.85 ± 2.80	7.32 ± 3.07	**9.55 ± 4.36 ****
		Serum	4.8 ± 3.09	4.97 ± 2.84	**11.98 ± 5.1 ****
		Erythrocytes	6.52 ± 3.05	8.017 ± 3.371	**11.44 ± 3.62 ****
	**F**	Whole blood	7.70 ± 3.31	7.90 ± 3.05	**10.30 ± 4.09 ***
		Serum	4.10 ± 3.10	5.30 ± 2.62	**13.75 ± 5.10 ***
		Erythrocytes	6.48 ± 3.23	7.50 ± 3.33	**10.54 ± 3.44 ****
	**M**	Whole blood	6.00 ± 2.29	6.50 ± 3.11	**7.50 ± 5.11 ****
		Serum	5.50 ± 3.08	4.50 ± 3.16	7.10 ± 5.10
		Erythrocytes	6.56 ± 2.87	8.75 ± 3.43	**13.91 ± 4.12 ****
**Nickel [µg/L]**	**All**	Whole blood	4.1 ± 1.81	**8.31 ± 3.05 ****	**11.37 ± 4.59 ****
		Serum	10.75 ± 3.81	10.03 ± 3.21	9.31 ± 3.49
		Erythrocytes	1 ± 0.345	**5.943 ± 2.618 ****	**14.70 ± 8.87 ****
	**F**	Whole blood	3.1 ± 1.37	**12.9 ± 4.33 ****	**13.8 ± 5.36 ****
		Serum	13.7 ± 4.23	15.7 ± 5.25	12.00 ± 4.31
		Erythrocytes	1.00 ± 0.30	**9.03 ± 3.45 ****	**16.29 ± 9.23 ****
	**M**	Whole blood	5.1 ± 2.24	**1.80 ± 1.23 ****	4.7 ± 2.47
		Serum	7.80 ± 3.38	**2.00 ± 0.31 ****	**1.90 ± 1.24 ****
		Erythrocytes	1.00 ± 0.39	1.57 ± 1.44	10.34 ± 7.88
**Cobalt [µg/L]**	**All**	Whole blood	6.96 ± 2.84	7.37 ± 3.91	6.79 ± 4.39
		Serum	1.20 ± 0.8	1.22 ± 0.64	1.42 ± 1.11
		Erythrocytes	15.68 ± 6.36	16.67 ± 6.42	15.62 ± 7.60
	**F**	Whole blood	7.02 ± 3.15	6.96 ± 3.33	6.78 ± 4.32
		Serum	1.10 ± 0.90	1.30 ± 0.60	1.50 ± 1.30
		Erythrocytes	16.71 ± 6.71	16.53 ± 5.99	16.04 ± 7.99
	**M**	Whole blood	6.90 ± 2.53	7.94 ± 4.73	6.81 ± 4.57
		Serum	1.30 ± 0.70	1.10 ± 0.70	1.20 ± 0.60
		Erythrocytes	14.64 ± 6.01	16.88 ± 7.02	14.47 ± 6.52
**Manganese [µg/L]**	**All**	Whole blood	42.95 ± 7.56	46.21 ± 8.34	40.95 ± 9.36
		Serum	14 ± 6.11	16.43 ± 6.71	**18.31 ± 7.22 ***
		Erythrocytes	95.82 ± 27.12	104 ± 43.5	93.69 ± 39.94
	**F**	Whole blood	42.1 ± 6.7	44.6 ± 8.3	41.2 ± 9.4
		Serum	16.6 ± 6.0	17.3 ± 7.0	18.1 ± 7.4
		Erythrocytes	99.5 ± 23.3	105.2 ± 44.5	97.1 ± 39.7
	**M**	Whole blood	43.8 ± 8.4	48.5 ± 8.4	40.2 ± 9.4
		Serum	11.4 ± 6.2	15.2 ± 6.2	**18.9 ± 6.8 ****
		Erythrocytes	92.1 ± 30.9	102.3 ± 42.1	84.5 ± 40.5
**Zinc [mg/L]**	**All**	Whole blood	6.447 ± 3.441	4.563 ± 2.911	4.902 ± 3.000
		Serum	3.544 ± 1.999	**1.641 ± 1.337 ****	**2.249 ± 1.507 ***
		Erythrocytes	6.447 ± 3.441	4.563 ± 2.911	4.902 ± 3.000
	**F**	Whole blood	6.788 ± 3.771	4.933 ± 2.921	4.780 ± 3.002
		Serum	3.513 ± 2.001	**1.752 ± 1.567 ****	**1.744 ± 1.211 ****
		Erythrocytes	11.31 ± 5.56	9.33 ± 6.45	8.97 ± 5.21
	**M**	Whole blood	6.106 ± 3.111	4.040 ± 2.899	5.239 ± 2.998
		Serum	3.575 ± 1.998	**1.484 ± 1.011 ****	3.637 ± 2.322
		Erythrocytes	8.96 ± 5.62	6.92 ± 4.45	7.05 ± 3.95

GFR—glomerular filtration rate, F—females, M—males, T2DM—type 2 diabetes mellitus. * The bolded results indicate statistically significant differences at levels *p* < 0.05. ** The bolded results indicate a highly statistically significant difference at levels *p* < 0.01. The remaining values (*p* > 0.05) indicate no statistically significant differences.

**Table 3 nutrients-16-02989-t003:** Elemental contents in the diet of diabetics and healthy people.

Trace Element	Control		T2DM with or without CKD		*p*
	Average	SD	Average	SD	
Zn	9.6	2.84	11.51	2.99	**<0.01 ***
Mn	4.16	1.48	6.29	1.72	**<0.01 ***
Ni	0.227	0.108	0.320	0.155	**<0.01 ***

Mn—manganese, Ni—nickel, Zn—zinc, T2DM—type 2 diabetes mellitus, SD—standard deviation. The *p*-values were assessed using the Mann–Whitney U test. * The bolded results indicate statistically significant differences.

## Data Availability

The original contributions presented in the study are included in the article/Appendix A, further inquiries can be directed to the corresponding author/s.

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
