# Peer review of "The Evaluation of Selected Trace Elements in Blood, Serum and Blood Cells of Type 2 Diabetes Patients with and without Renal Disorder"

_nutrients, 2024, doi:10.3390/nu16172989_

Round 1

Reviewer 1 Report

Comments and Suggestions for Authors

This manuscript represents an examination of circulating metal concentrations correlated with Type 3 Diabetes mellitus with or without chronic renal disorder. I have comments for minor revision to be addressed by the authors in order to render the manuscript publishable.

1. Line 42: The word "changes" should not be in this sentence. Lifestyle habits including eating habits contribute to the disease. Lifestyle changes are needed to improve patient outcomes from the disease.

2. Lines 118-122, 155-157, and lines 267-269: What kind of tubes were used during the digestion in nitric acid? It is well known that metals are leached by acids from glass. If glass tubes were used, it may explain some of the variability in the data, and the lack of differences in cobalt concentrations between control and diseased patients. If cobalt leached from glass overwhelmed the blood concentration, this would explain it. If glass tubes were used, it may also explain the slightly high results for zinc in Table 1. The authors need to learn appropriate analytical methodological practices and descriptions of the methods in manuscripts when publishing analytical results. I hope that either PFA or low metals background polypropylene tubes were used. The authors must state in lines 118-122 what type of tubes were used.

3. Lines 119-121: All results are dependent on analytical chemistry, therefore the authors must be more detailed in methodological descriptions, including the type of tubes used as stated above. How were the samples prepared for analysis after "mineralization"? Were they diluted with water from a water purification system? To what volume or with what dilution factor were they diluted?

4. Linea 129-130: The authors state that reference materials from Merck were used, but this is insufficient information. What reference materials were used? Were they diluted into nitric acid at same concentrations as the samples prepared after mineralization? Were the sample concentrations bracketed by standard concentrations? What kind of blank was subtracted from standards used for calibration? from samples?

5. Lines 205-206, 232-234, 267-268, 277: The authors seem surprised that several metal concentrations are higher in whole blood or red blood cells than in serum. It is common knowledge that blood transition metals are predominantly cell associated and lower as "free" metals in serum.

6. Lines 260-261: Nickel compounds absorb into the epidermal layer of skin sufficiently to cause epidermal nickel allergies but not sufficiently to cause changes in circulating nickel concentrations. The statement about advising against wearing nickel jewelry should be deleted.

7. Lines 373-374: Correct broken sentences in these lines.

8. Lines 447-451: Could the correlation between cobalt and health be due to vitamin B12 (methylcobalamin) be the reason that there is an association between cobalt status and health but adding cobalt to diet does not help? Unless cobalt is in the form of methylcobalamin, I am not surprised the cobalt supplementation did not help humans.

9. Lines 459-461: The authors need to be careful what they are recommending. Methylcobalamin is the principal form of cobalt known to be nutritionally useful . Cobalt compounds in the cobalt(II) oxidation state are IARC group 2A and 2B carcinogens. The statement in these lines must be more specific in order to avoid recommending carcinogen supplementation.

Comments on the Quality of English Language

Most of the text is fine. There are a few broken sentences and a statement in the first paragraph of Introduction that needs correction as described above.

Author Response

Thank the Reviewer’s very much for your time, profound analysis and valuable comments on our manuscript. The responses for all points are below. The changes were introduced into the text of manuscript, as suggested by the Reviewer.

Point 1: Line 42: The word "changes" should not be in this sentence. Lifestyle habits including eating habits contribute to the disease. Lifestyle changes are needed to improve patient outcomes from the disease.

Response 1: Thank the Reviewer’s suggestion. We changed the word “changes” to “lifestyle properties” in line 42.

Point 2:  Lines 118-122, 155-157, and lines 267-269: What kind of tubes were used during the digestion in nitric acid? It is well known that metals are leached by acids from glass. If glass tubes were used, it may explain some of the variability in the data, and the lack of differences in cobalt concentrations between control and diseased patients. If cobalt leached from glass overwhelmed the blood concentration, this would explain it. If glass tubes were used, it may also explain the slightly high results for zinc in Table 1. The authors need to learn appropriate analytical methodological practices and descriptions of the methods in manuscripts when publishing analytical results. I hope that either PFA or low metals background polypropylene tubes were used. The authors must state in lines 118-122 what type of tubes were used.

Response 2: Thank the Reviewer’s suggestion. Only some people are aware of this information, so we used special polypropylene tubes coated with lithium heparin (https://www.sarstedt.com/en/products/diagnostic/venous-blood/s-monovette/product/01.1604.400/). We added the statement that used tubes were made of polypropylene in line 111.

Point 3: Lines 119-121: All results are dependent on analytical chemistry, therefore the authors must be more detailed in methodological descriptions, including the type of tubes used as stated above. How were the samples prepared for analysis after "mineralization"? Were they diluted with water from a water purification system? To what volume or with what dilution factor were they diluted?

Response 3: Thank the Reviewer’s suggestion. The samples were collected into Monovette (Sarstedt) closed-system tubes with lithium-heparin coating. Paralell to samples, a standard reference serum (Seronorm) was transferred manualny to Monovette tubes and processed with samples. After mieralization in 65%  Merck Millipore Suprapur nitric(V) acid, the samples were diluted to a Total volume of 25 ml with hplc-grade water. We added adequate information’s in manuscript lines 110-114.

Point 4: Lines 129-130: The authors state that reference materials from Merck were used, but this is insufficient information. What reference materials were used? Were they diluted into nitric acid at same concentrations as the samples prepared after mineralization? Were the sample concentrations bracketed by standard concentrations? What kind of blank was subtracted from standards used for calibration? from samples?

Response 4: Thank the Reviewer’s suggestion. Standards from Merck were AAS standards for respective metal ions, diluted into the same batch of Suprapur nitric acid (Merck). Blank sample was prepared with hplc-grade water in place of serum, then was mineralized and diluted along with the samples. The AAS  rresult of this sample was deducted from other results to correct for the chemical impurities in the chemical reagents used. We added adequate information’s in manuscript lines 137-141.

Point 5: Lines 205-206, 232-234, 267-268, 277: The authors seem surprised that several metal concentrations are higher in whole blood or red blood cells than in serum. It is common knowledge that blood transition metals are predominantly cell associated and lower as "free" metals in serum.

Response 5: Thank the Reviewer’s suggestion. We added the phrase as the Reviewer proposed in Discussion paragraph in lines 416-418 to explain results obtained in trace elements from serum and blood cells .

Point 6. Lines 260-261: Nickel compounds absorb into the epidermal layer of skin sufficiently to cause epidermal nickel allergies but not sufficiently to cause changes in circulating nickel concentrations. The statement about advising against wearing nickel jewelry should be deleted.

Response 6: Thank the Reviewer’s suggestion. We deleted the statements about avoiding wearing nickel jewelry.

Point 7: Lines 373-374: Correct broken sentences in these lines.

Response 7: Thank the Reviewer’s suggestion. We corrected the phrases as the Reviewer requested in lines 373-374.

Point 8: Lines 447-451: Could the correlation between cobalt and health be due to vitamin B12 (methylcobalamin) be the reason that there is an association between cobalt status and health but adding cobalt to diet does not help? Unless cobalt is in the form of methylcobalamin, I am not surprised the cobalt supplementation did not help humans.

Response 8: Thank the Reviewer’s suggestion. We added the information about methylcobalamin in lines 487-489.

Point 9: Lines 459-461: The authors need to be careful what they are recommending. Methylcobalamin is the principal form of cobalt known to be nutritionally useful . Cobalt compounds in the cobalt(II) oxidation state are IARC group 2A and 2B carcinogens. The statement in these lines must be more specific in order to avoid recommending carcinogen supplementation.

Response 9: Thank the Reviewer’s suggestion. We added the information about cobalt carcinogenic properties and corrected our recommendations in lines 500-504.

We sincerely hope that all changes introduced by us in the text will be fully satisfactory for the Reviewer.

Reviewer 2 Report

Comments and Suggestions for Authors

In this manuscript (ID# nutrients-3166742) entitled “The evaluation of selected trace elements in blood, serum and blood cells of type 2 diabetes patients with and without renal disorder”, authors Frankowski et al have studied plasma mineral levels in patients with type 2 diabetes with/without renal impairment as compared with healthy individuals. Their results have demonstrated that the plasma mineral levels are altered in those patients. They conclude that the standard diet including those minerals should be modified in those patients. However, there are several major concerns, which are listed in the following paragraphs:

1. Please clarify the standard of healthy subjects in the current study, because many other chronic diseases could also alter plasma mineral levels in addition to renal failure and diabetes.

2. In Table 1, the first line “Nickel μg/l] Zinc [mg/l]” should be modified. The content of Table 2 should be delated and the information in this table could be added to the Methods.

3. In table 3, the data should be statistic analyzed and the statistical significance test should be labeled in the table.

4. It would be better to combine significance test and mean plasma content together in one table instead of listing the significance tests as separated tables (tables 14, 15).

5. How elemental contents in the daily diets were measured in Table 16? Why the plasma mineral levels in patients with diabetes and/or renal impairment are altered? Due to reduced dietary supply, decreased absorption, or altered excretion in kidney?

Comments on the Quality of English Language

The overall quality of English language should be improved.

Author Response

Thank the Reviewer’s very much for your time, profound analysis and valuable comments on our manuscript. The responses for all points are below. The changes were introduced into the text of manuscript, as suggested by the Reviewer.

Point 1:. Please clarify the standard of healthy subjects in the current study, because many other chronic diseases could also alter plasma mineral levels in addition to renal failure and diabetes.

Response 1: Thank the Reviewer’s suggestion. We clarified the standards of control group and added more specific inclusion/exclusion criteria’s in lines 75-81.

Point 2: In Table 1, the first line “Nickel μg/l] Zinc [mg/l]” should be modified. The content of Table 2 should be delated and the information in this table could be added to the Methods.

Response 2: Thank the Reviewer’s suggestion. We added the “[“ in Nickel concentrations in Table 1.

Point 3: In table 3, the data should be statistic analyzed and the statistical significance test should be labeled in the table.

Response 3: Thank you very much for this comment. As suggested, we have provided in Table 3 (now Table S1) the levels of statistical significance between the compared patient groups.

Point 4: It would be better to combine significance test and mean plasma content together in one table instead of listing the significance tests as separated tables (tables 14, 15).

Response 4: Thank the Reviewer’s suggestion. We deleted tables 14 and 15 from manuscript and attached them as a supplementary materials, also the information about significance were added to adequate tables in manuscript.

Point 5: How elemental contents in the daily diets were measured in Table 16? Why the plasma mineral levels in patients with diabetes and/or renal impairment are altered? Due to reduced dietary supply, decreased absorption, or altered excretion in kidney?

Response 5: Thank the Reviewer’s suggestion. The element contents were calcuated basen on nutrition value tables and/or determined with AAS method. The patients' diets were controlled for three days prior to sample collection. However, the design of this experient does not explain the variability of the element content.

Point 6: The overall quality of English language should be improved.

Response 6: Thank the Reviewer’s suggestion. We performed a careful reading of the manuscript to improve punctuation marks and general look of the English quality. We also corrected spellings throughout the manuscript.

We sincerely hope that all changes introduced by us in the text will be fully satisfactory for the Reviewer.

Round 2

Reviewer 2 Report

Comments and Suggestions for Authors

The manuscript has been improved. No further recommendation. 

Author Response

Thank the Reviewer very much for your time, profound analysis and valuable comments on our manuscript.